# Joule spectroscopy of hybrid superconductor–semiconductor nanodevices

A. Ibabe[1,2,5], M. Gómez[1,2,5], G. O. Steffensen[2,3], T. Kanne[4], J. Nygård[4], A. Levy Yeyati[2,3] & E. J. H. Lee[1,2] ✉

Hybrid superconductor-semiconductor devices offer highly tunable platforms, potentially suitable for quantum technology applications, that have been intensively studied in the past decade. Here we establish that measurements of the superconductor-to-normal transition originating from Joule heating provide a powerful spectroscopical tool to characterize such hybrid devices. Concretely, we apply this technique to junctions in full-shell Al-InAs nanowires in the Little-Parks regime and obtain detailed information of each lead independently and in a single measurement, including differences in the superconducting coherence lengths of the leads, inhomogeneous covering of the epitaxial shell, and the inverse superconducting proximity effect; all-in-all constituting a unique fingerprint of each device with applications in the interpretation of low-bias data, the optimization of device geometries, and the uncovering of disorder in these systems. Besides the practical uses, our work also underscores the importance of heating in hybrid devices, an effect that is often overlooked.

The possibility to generate topological superconductivity in hybrid superconductor-semiconductor nanostructures[1–3] has driven a strong interest towards this material platform in the past decade. Recent work has also targeted the development of novel quantum devices using the same combination of materials in the trivial regime[4–9]. In spite of the remarkable developments in crystal growth and fabrication in recent years[10–13], material and device imperfections remain important outstanding challenges for the above research directions. Indeed, it is now generally accepted that disorder constitutes the main hurdle for the realization of a topological phase in hybrid nanowires[14,15] and, consequently, for the development of a topological qubit. Clearly, further improvements in the quality of crystals are crucial for advancing the field. In parallel, there is also a need for characterization tools that enable to efficiently probe the properties of the above materials, which is essential for identifying

sources of imperfections and for understanding at depth the response of fabricated devices[16]. In this work, we show that the Joule effect can be used as the basis for such a characterization tool for hybrid superconducting devices[17,18]. We demonstrate the potential of the technique by studying devices based on full-shell Al-InAs nanowires, also in the Little-Parks regime[19], and uncover clear signatures of disorder and defects in the epitaxial shell, as well as device asymmetries resulting from the inverse superconducting proximity effect from normal metal contacts. Our results emphasize the high degree of variability present in this type of system, as well as the importance of heating effects in hybrid devices.

The Joule effect describes the heat dissipated by a resistor when an electrical current flows, with a corresponding power equal to the product of the current and voltage in the resistor, $P = VI$. While Joule heating in superconducting devices is absent when the electrical

¹Departamento de Física de la Materia Condensada, Universidad Autónoma de Madrid, Madrid, Spain. ²Condensed Matter Physics Center (IFIMAC), Universidad Autónoma de Madrid, Madrid, Spain. ³Departamento de Física Teórica de la Materia Condensada, Universidad Autónoma de Madrid, Madrid, Spain. ⁴Center for Quantum Devices, Niels Bohr Institute, University of Copenhagen, Copenhagen, Denmark. ⁵These authors contributed equally: A. Ibabe, M. Gómez. ✉e-mail: eduardo.lee@uam.es

current is carried by Cooper pairs, it reemerges when transport is mediated by quasiparticles. Interestingly, owing to the intrinsically poor thermal conductivity of superconductors at low temperatures, heating effects can be further amplified by the formation of bottlenecks for heat diffusion. As a result, the Joule effect can have a strong impact on the response of such devices. Indeed, heating has been identified as the culprit for the hysteretic $I$–$V$ characteristics of superconducting nanowires (NWs)[20] and overdamped $S$–$N$–$S$ Josephson junctions (where $S$ and $N$ stand for superconductor and normal metal, respectively)[21], as well as for missing Shapiro steps in the latter[22]. In addition, it has been shown that the injection of hot electrons can significantly impact the Josephson effect in metallic[23] and in InAs NW-based devices[24], ultimately leading to the full suppression of the supercurrent for sufficiently high injected power.

Here, we show that instead of being merely a nuisance, Joule heating can also provide rich and independent information regarding each superconducting lead in hybrid superconductor-semiconductor devices in a single measurement, which can be put together to obtain a device fingerprint. To this end, we follow previous work on graphene-based Josephson junctions (JJs)[17,18] and study the Joule-driven superconductor-to-normal metal transition of the leads in nanowire devices. Such a transition yields a clear signature in transport, namely a finite bias dip in the differential conductance, d$I$/d$V$, which can be used for performing spectroscopical-type measurements of the superconductivity

of the leads at low temperatures. Importantly, we demonstrate that this technique, which we dub Joule spectroscopy, is able to bring to light very fine details that would otherwise be difficult to obtain only from the low-bias transport response, thus underscoring its potential for the characterization of hybrid superconducting devices. To demonstrate the technique, we focus on devices based on full-shell epitaxial Al-InAs nanowires. Specifically, we study JJs obtained by wet etching a segment of the Al shell, as schematically shown in Fig. 1a for device A (see Methods for a detailed description of the fabrication and of the different devices). An electron micrograph of a typical device is shown in Supplementary Information Fig. S1. For reasons that will become clearer later, we note that the leads in our JJs can display different values of superconducting critical temperature, $T_{c,i}$, and gap, $\Delta_i$, where $i$ refers to lead 1 or 2.

## Results

### Principle of Joule spectroscopy

We start by addressing the working principle of Joule spectroscopy in greater detail. The technique relies on the balance between the Joule heat dissipated across the junction of a hybrid device and the different cooling processes, such as electron-phonon coupling and quasiparticle heat diffusion through the leads. As both cooling processes become inefficient at low temperatures[25–27], a heat bottleneck is established and the temperature around the junction increases (Fig. 1a). Here, we neglect cooling by electron-phonon coupling as we estimate it to be weak (see Supplementary information (SI)). We now turn to the impact of the Joule heating on the transport response of the devices. In Fig. 1b, we plot $I(V)$ and d$I$/d$V(V)$ traces for device A. The observed low-bias response is typical for JJs based on semiconductor nanostructures. We ascribe the d$I$/d$V$ peaks in this regime to a Josephson current at $V = 0$ and multiple Andreev reflection (MAR) resonances at $V = 2\Delta/ne$ where, for this device, $\Delta = \Delta_1 = \Delta_2 \approx 210\ \mu$eV. Moreover, for $V \geq 2\Delta/e$, the $I$–$V$ curve is well described by the relation,

$$I = V/R_J + I_{exs,1}(T_{0,1}) + I_{exs,2}(T_{0,2}), \qquad (1)$$

where $R_J$ is the normal state junction resistance and $I_{exs,i}(T_{0,i})$ is the excess current resulting from Andreev reflections at lead $i$. Crucially, the excess current depends on the temperature of the leads at the junction, $T_{0,i}$, which can differ from each other owing to device asymmetries. For $V \lesssim 2.5$ mV, the $I_{exs,i}$ terms are approximately constant, leading to a linear $I$–$V$ characteristic. However, as Joule heating intensifies, deviations from this linear response follow the suppression of the excess current as $T_{0,i}$ approaches $T_{c,i}$, and $\Delta_i$ closes. At a critical voltage $V = V_{dip,i}$, the lead turns normal ($T_{0,i} = T_{c,i}$), and the excess current is fully suppressed (red dashed line in Fig. 1b), giving rise to dips in d$I$/d$V$[7,18]. We show in the following that such dips can be used for a detailed characterization of the devices.

To this end, we model the system as an $S$–$S$ junction with $N$ conduction channels of transmission $\tau$ connecting the two superconducting leads[28]. We further assume that injected electrons and holes equilibrate to a thermal distribution within a small distance of the junction. This is supported by the short mean-free path of the Al shell, $l \sim$ nm (see SI for an estimate in our devices)[29], compared to the typical length of the leads, $L \sim \mu$m. This equilibration results in a power, $P_i$, being deposited on either junction interface, which propagates down the Al shell by activated quasiparticles as depicted in Fig. 1a and c. By solving the resulting heat diffusion equation at $T_{0,i} = T_{c,i}$, whereby we assume that the other end of the Al shell is anchored at the bath temperature of the cryostast, $T_{bath}$, we obtain a metallic-like Wiedemann-Franz relation for the critical power at

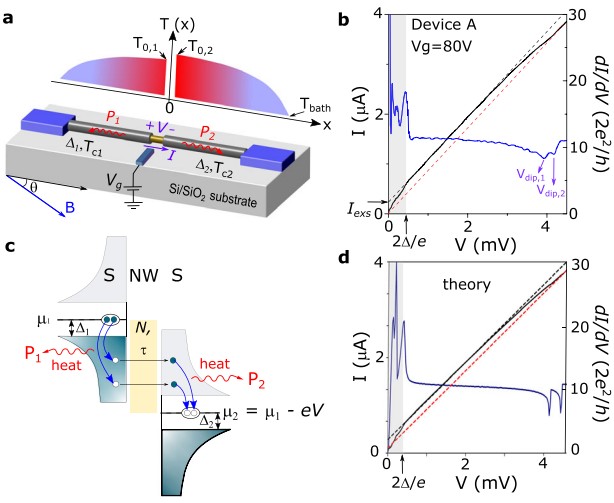

**Fig. 1 | Principle of Joule spectroscopy. a** Schematics of the device geometry. A Josephson junction is formed by etching a 200-nm segment of a full-shell Al-InAs nanowire (NW). Voltage applied to a side gate, $V_g$, tunes the junction resistance, $R_J$. The balance between the Joule heat dissipated at the nanowire junction (equal to the product of the voltage, $V$, and current, $I$) and the cooling power from the superconducting leads 1 and 2 ($P_1$ and $P_2$) results in a temperature gradient along the device, $T(x)$. At a critical value of Joule dissipation, the temperature of the leads, $T_{0,1}$ and $T_{0,2}$, exceed the superconducting critical temperature and the leads turn normal. Each lead can display different superconducting gaps $\Delta_1$ and $\Delta_2$. An external magnetic field, $B$, is applied with an angle $\theta$ to the NW axis. $T_{bath}$ is the cryostat temperature. **b** $I$ (solid black line) and differential conductance, d$I$/d$V$ (solid blue line), as a function of $V$ measured at $V_g = 80$ V in device A. For $V < 2\Delta/e$, transport is dominated by Josephson and Andreev processes. By extrapolating the $I$–$V$ curve just above $V = 2\Delta/e$, an excess current of $I_{exs} \approx 200$ nA is estimated (dashed black line). Upon further increasing $V$, the Joule-mediated transition of the superconducting leads to the normal state manifests as two d$I$/d$V$ dips ($V_{dip,1}$ and $V_{dip,2}$). These transitions fully suppress $I_{exs}$ (dashed red line). **c** The nanowire is modeled as a quasi-ballistic conductor with $N$ conduction channels with transmissions $\tau$. We assume that the energy of the quasiparticles injected in the superconductors is fully converted into heat. **d** Keldysh-Floquet calculations of $I(V)$ and d$I$/d$V(V)$ using device A parameters (see Supplementary information for more information), reproducing the main features in (**b**).

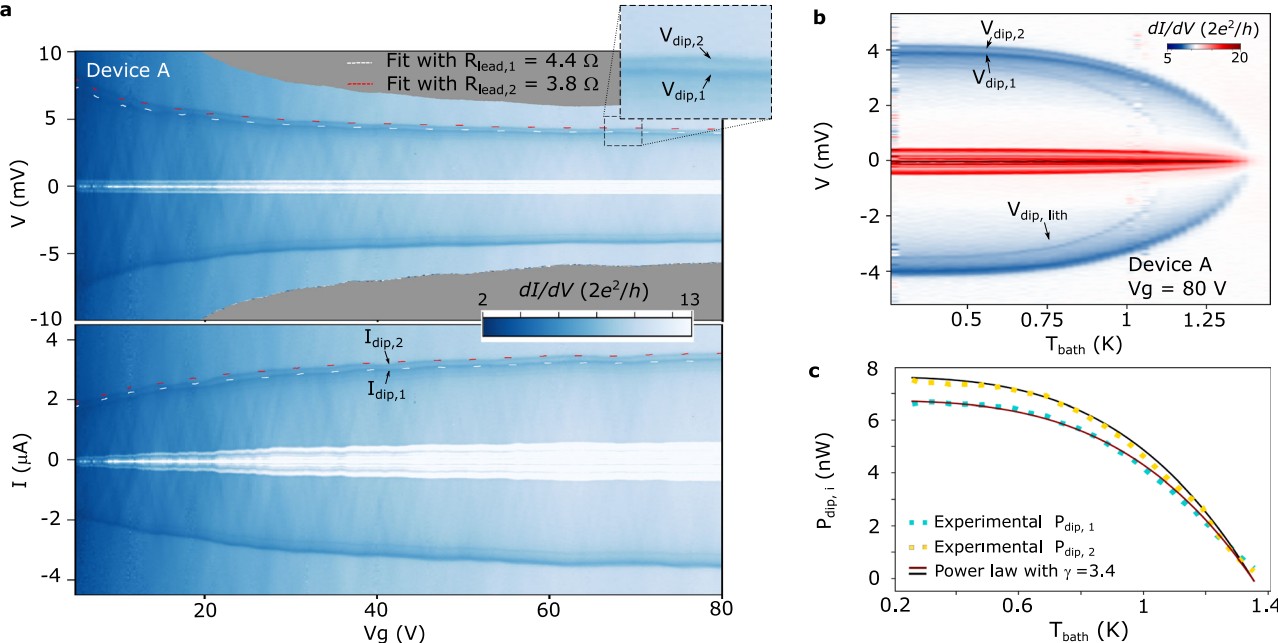

**Fig. 2 | Characterization of the superconductor-to-normal metal transition of the epitaxial Al leads. a** Gate voltage dependence of the d$I$/d$V$ for device A. The data is plotted both as a function of $V$ (top panel) and of $I$ (bottom panel). Enhanced d$I$/d$V$ features at low $V$ and $I$ can be attributed to Josephson and Andreev processes. Two d$I$/d$V$ dips, which signal the superconductor-to-normal metal transition of the leads, can be identified in each of the panels ($V_{dip,i}$ and $I_{dip,i}$). The presence of the two dips is shown in greater detail in the inset of the top panel. The white and red dashed lines are fits to Eq. (3) with a single free fitting parameter per lead ($R_{lead,1}$ and $R_{lead,2}$). **b** d$I$/d$V$ as a function of $V$ and of $T_{bath}$. A faint dip at $V_{dip,lith}$ is attributed to the Ti/Al contacts to the NW. **c** $P_{dip,1} = V_{dip,1}I_{dip,1}/2$ (blue squares) and $P_{dip,2} = V_{dip,2}I_{dip,2}/2$ (yellow squares) as a function of $T_{bath}$. The solid lines are fits to the power law in Eq. (4), yielding an exponent $\gamma = 3.4$.

which the dips appear (see SI),

$$P_{dip,i} = \Lambda \frac{k_B^2 T_{c,i}^2}{e^2 R_{lead,i}}, \quad (2)$$

where $R_{lead,i}$ is the normal resistance of the leads, and $\Lambda$ accounts for details of heat diffusion, which for the majority of experimental parameters is approximately equal to the zero-temperature BCS limit, $\Lambda \approx 2.112$ (see SI for a detailed discussion). In the high-bias limit at which the dips appear, the ohmic contribution to the current dominates $V/R_J \gg I_{exs,i}(T_{0i})$, and consequently $P_1 \approx P_2 \approx IV/2 \approx V^2/2R_J$, which implies

$$V_{dip,i} = R_J I_{dip,i} = \sqrt{2\Lambda} \sqrt{\frac{R_J}{R_{lead,i}}} \frac{k_B T_{c,i}}{e}, \quad (3)$$

where $I_{dip,i}$ is the current value for the dips. Equation (2) and Eq. (3) constitute the main theoretical insights of this work and establish the basis for Joule spectroscopy. Indeed, the direct relation between $I_{dip,i}$ and $V_{dip,i}$ to $T_{c,i}$ reveals how measurements of the dips can be used to probe the superconducting properties of the leads. To support these relations we calculate $I$ and $P_i$ self-consistently in $T_{0,i}$ by using the Floquet–Keldysh Green function technique. This allows us to compare low-bias MAR structure with high-bias dip positions, and include effects of varying $\Lambda$, finite $I_{exs,i}(T_{0,i})$, pair-breaking, $\alpha$, from finite magnetic fields, and the influence of lead asymmetry on transport. Results of these calculations are shown in Fig. 1d and later figures with additional details given in the Supplementary Information.

To confirm the validity of our model, we study the dependence of the dips on $R_J$, which is tuned by electrostatic gating. Following Eq. (3), we expect $V_{dip,i}$ ($I_{dip,i}$) to be directly (inversely) proportional to $\sqrt{R_J}$. Figure 2a displays d$I$/d$V(V)$ (top panel) and d$I$/d$V(I)$ (bottom panel) of device A as a function of gate voltage, $V_g$. Within the studied $V_g$ range, $R_J$ varies by a factor of ~4. In analogy to Fig. 1b, the high conductance regions for low $V$ ($V < 2\Delta/e$) and $I$ are due to Josephson and Andreev

transport. For $V$ well above the gap, a pair of d$I$/d$V$ dips are detected at $V_{dip,i}$ and $I_{dip,i}$. As shown in the inset of Fig. 2a, the two dips are better resolved for positive $V$ ($I$), reflecting a small asymmetry with respect to the sign of the bias. We fit the positions of the dips to Eq. (3) using $R_{lead,i}$ as a single free fitting parameter per lead/dip, as well as the experimental values for $R_J$ and $T_c = T_{c,1} = T_{c,2} = 1.35$ K. The fits, shown as white and red dashed lines in Fig. 2a, agree remarkably well with the experimental data, thus strongly supporting our model. From these, we obtain $R_{lead,1} = 4.4\ \Omega$ and $R_{lead,2} = 3.8\ \Omega$, consistent with the normal state resistance of the Al shell (~10$\Omega$/$\mu$m, as measured in nominally identical NWs (see SI)) and lead lengths $L_i \sim 0.5\ \mu$m. The different values of $R_{lead,i}$ are attributed to slight device asymmetries, e.g., differences in $L_i$. Note that the good agreement of both $V_{dip,i}$ and $I_{dip,i}$ to the model demonstrates that $P_{dip,i}$ is independent of $R_J$, as expected from Eq. (2)[18].

Further information about the dips is gained by studying their dependence on $T_{bath}$. As shown in Fig. 2b, both $V_{dip,1}$ and $V_{dip,2}$ go to zero at $T_{bath} = T_c \approx 1.35$ K, underscoring their superconductivity-related origin. Interestingly, an additional pair of faint d$I$/d$V$ dips with a lower critical temperature of $T_{c,lith} \approx 1.1$ K is observed. We conclude that these faint dips are related to the superconductivity of the lithographically defined Al contacts shown in blue in Fig. 1a (Supplementary information to: Joule spectroscopy of hybrid superconductor-semiconductor nanodevices (2022).). The $T_{bath}$-dependence of the dips can also provide insights regarding the heat dissipation mechanisms of the device. As shown in Fig. 2c, the critical power of the dips can be fitted to

$$\frac{P_{dip,i}(T_{bath})}{P_{dip,i}(T_{bath}=0)} = 1 - \left(\frac{T_{bath}}{T_{c,i}}\right)^\gamma, \quad (4)$$

yielding $\gamma \approx 3.4$. Note that there are no additional fitting parameters to the curves and that $P_{dip,i}(T_{bath}=0)$ is calculated from the experimental $R_J$ and $R_{lead,i}$ extracted from the fits in Fig. 2a. As shown in the Supplementary Information, we numerically calculate $P_{dip}$ as a function of $T_{bath}$ and fit the resulting curve to eq. (4), obtaining $\gamma^{theory} \approx 3.6$, which is

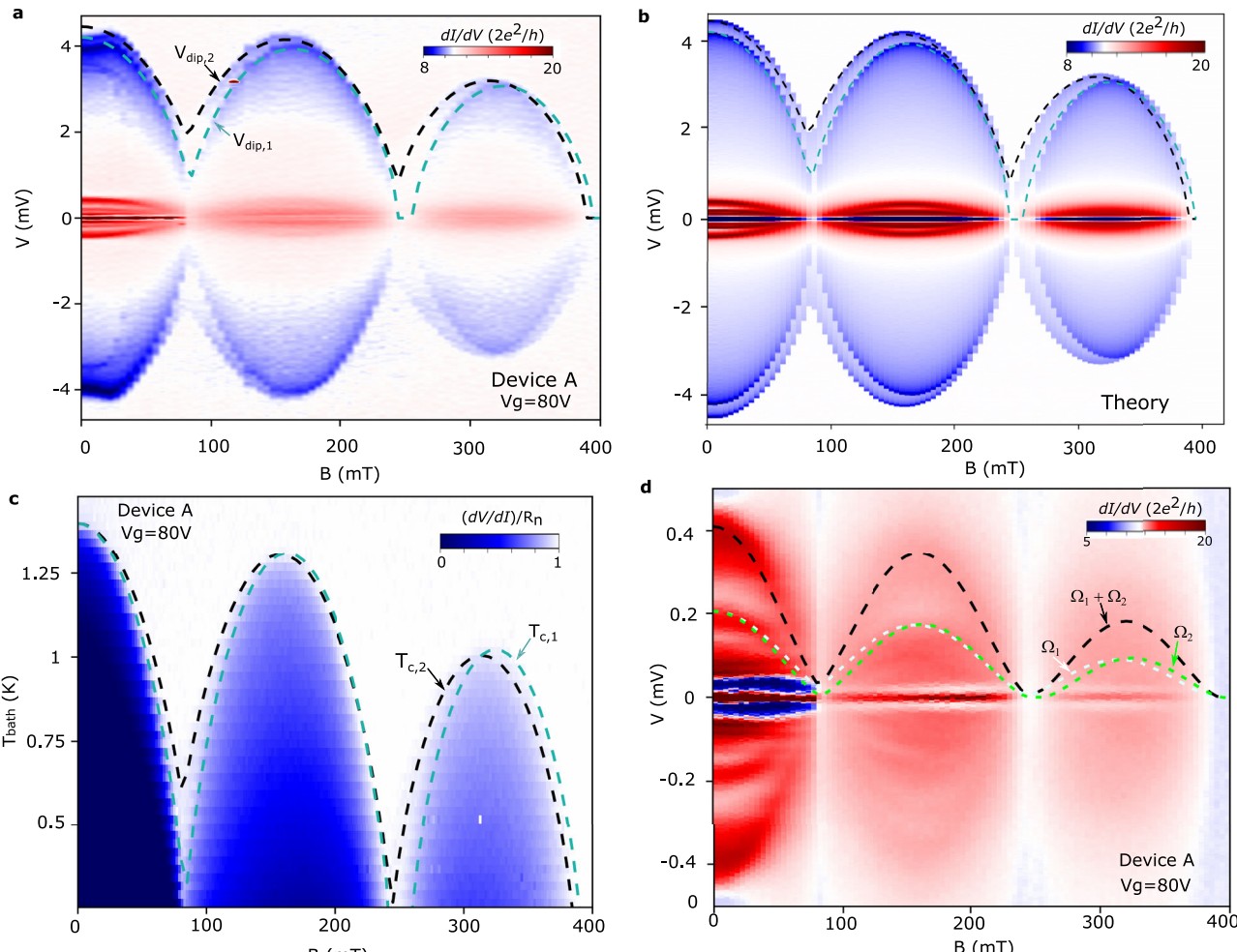

**Fig. 3 | Joule effect as a spectroscopical tool. a** Oscillations of $V_{dip,1}$ and $V_{dip,2}$ with applied magnetic field, which result from the modulation of $T_{c,i}$ by the Little Parks (LP) effect. The dashed lines are fits to the Abrikosov–Gor'kov (AG) theory, from which we conclude that the primary cause for the different LP oscillations are differences in the superconducting coherence lengths of the leads.
**b** Keldysh–Floquet calculations of the Andreev conductance at low $V$ and of the d$I$/d$V$ dips at high $V$ as a function of $B$ using device A parameters (Supplementary information to: Joule spectroscopy of hybrid superconductor-semiconductor nanodevices (2022).), capturing the main experimental observations. Panels (**c**) and (**d**) demonstrate the spectroscopical potential of the technique. **c** Zero-bias d$V$/d$I$ normalized by the normal state resistance of the device. The dashed lines correspond to $T_{c,i}(B)$ calculated with the AG parameters extracted by fitting the dips in panel (**a**). **d** Low-$V$ transport characterization of device A as a function of $B$. The dashed lines show the spectral gaps, $\Omega_1(B)/e$ (white) and $\Omega_2(B)/e$ (green), and their sum, $(\Omega_1(B) + \Omega_2(B))/e$ (black), obtained from $V_{dip,i}(B)$.

in excellent agreement with our experimental results. This supports our assumption that quasiparticle heat diffusion constitutes the dominant cooling mechanism in our devices.

## Obtaining a device fingerprint

We now address the potential of Joule heating as a spectroscopical tool for hybrid superconducting devices. To accomplish this, we fix $R_J$ and study how the dips evolve as $T_{c,i}$ is tuned by an external magnetic field, $B$, approximately aligned to the NW axis (Fig. 1a). Figure 3 displays such a measurement for device A, taken at $V_g = 80$ V. Clear oscillations of $V_{dip,i}$ are observed, reflecting the modulation of $T_{c,i}$ with applied magnetic flux by the Little-Parks effect[19,30–32]. Surprisingly, the dips exhibit different Little-Parks oscillations, suggesting that the $T_{c,i}(B)$ dependences of the two leads are not the same. To clarify this, we employ the Abrikosov–Gor'kov (AG) theory[33,34] to fit the experimental data (dashed lines in Fig. 3a, see Methods for more information). Note that the good agreement between the dips and AG fitting is already a first indication that $V_{dip,i}$ and $T_{c,i}$ are approximately proportional, which is a consequence of $\Lambda$ remaining nearly constant within the experimental parameter space. The discrepancies at low $B$ can be

attributed to the lithographically-defined Al contacts, as we discuss in SI. The AG fitting additionally reveals that the distinct dip oscillations primarily result from differences in the superconducting coherence lengths of the leads, $\xi_{S,1} \approx 100$ nm and $\xi_{S,2} \approx 90$ nm, which owes to disorder in the epitaxial Al shell (for superconductors in the dirty limit, $\xi_S \propto \sqrt{l_e}$, where $l_e$ is the mean free path)[29], (see SI for more details). The main features of the experimental data are well captured by the results of our Floquet–Keldysh calculations using parameters obtained from the AG fitting (Fig. 3b).

Further support for Joule spectroscopy is gained by verifying that $V_{dip,i}$ and $T_{c,i}$ remain proportional as a function of $B$. To this end, we measure the differential resistance, d$V$/d$I$, of the device at $V = 0$, as shown in Fig. 3c. Regions in which d$V$/d$I$ < $R_n$, where $R_n$ is the normal state resistance, indicate that at least one of the leads is superconducting, whereupon the device conductance is enhanced either by Josephson or Andreev processes. The dashed lines correspond to the expected values of $T_{c,i}(B)$ from AG theory, which were calculated from the experimental zero-field critical temperature ($T_c = 1.35$ K) and parameters obtained from AG fitting in Fig. 3a. A very good agreement with the experimental data is observed, also allowing to identify

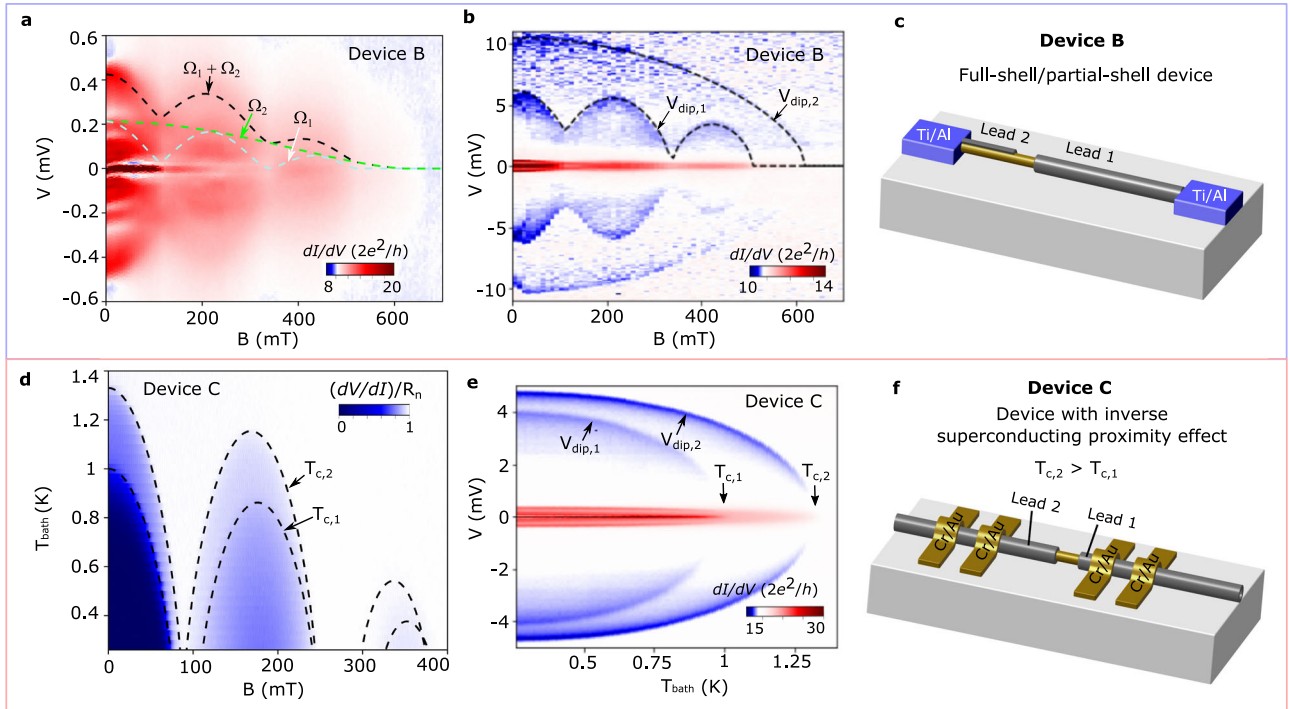

**Fig. 4 | Application of Joule spectroscopy to different NW devices. a** Low-bias transport characterization of device B as a function of magnetic field. Dashed lines show fittings of the spectral gaps, $\Omega_1(B)/e$ (white) and $\Omega_2(B)/e$ (green), and their sum, $(\Omega_1(B) + \Omega_2(B))/e$ (black), obtained from $V_{dip,i}(B)$. **b** Joule spectroscopy as a function of $B$ clearly identifies that one of the superconducting leads is not doubly-connected, i.e., it behaves as a partial-shell lead. Dashed lines are fits to the AG theory. **c** Schematics of device B, as concluded from the Joule spectroscopy char-acterization (not to scale). **d** $(dV/dI)/R_n$ as a function of $T$ and $B$ for device C. The dashed lines correspond to $T_{c,i}$ obtained from $T_{c,i}(B = 0)$ and the AG fits to $V_{dip,i}(B)$ (not shown, see SI ). **e** $T$-dependence of $V_{dip,1}$ and $V_{dip,2}$ in device C. Lead 1 displays a lower critical temperature owing to its closer proximity to the lithographic Cr/Au contacts, as depicted in the schematics in panel **f** (not to scale).

regions in which only one of the leads is superconducting (i.e., between the dashed lines, where $dV/dI$ takes values slightly lower than $R_n$). This demonstrates that the linear relation between $V_{dip,i}$ and $T_{c,i}$ is preserved for experimentally-relevant conditions, as required by the technique. We also stress that while the differences in $\xi_{S,i}$ are barely visible in Fig. 3c, they can be detected in a significantly clearer (and faster) manner using Joule spectroscopy. Overall, the above observa-tions demonstrate the ability of the technique in obtaining a device fingerprint. We emphasize that such detailed information of the superconducting leads separately is not directly accessible from the low-bias transport response, which we discuss below.

We now show that the information gained from Joule spectro-scopy provides a consistent description of the low-bias device response with respect to the experimental data (Fig. 3d). For this comparison, we focus on MAR resonances of orders $n = 1$ and 2 which, for $B = 0$, are centered at $V = (\Delta_1 + \Delta_2)/e$, and $V = \Delta_1/e$ and $V = \Delta_2/e$, respectively ($\Delta_i$ are obtained from the experimental $T_{c,i}$ using the BCS relation $\Delta \approx 1.76 k_B T_c$ valid at zero field). Owing to depairing effects, the MAR resonances cease to depend linearly on $\Delta_i$ and $T_{c,i}$ at finite $B$. Instead, the position of MAR peaks is better captured by scalings with the spectral gap, $\Omega_i(B) = \Delta_i(B = 0)(T_{c,i}(B)/T_{c,i}(B = 0))^{5/2}$, as concluded from our numer-ical simulations (see SI). In Fig. 3d, we plot $(\Omega_1 + \Omega_2)/e$ (black), $\Omega_1/e$ (white), and $\Omega_2/e$ (green) as dashed lines, which were calculated using $T_{c,i}(B)$ extracted from the dips in Fig. 3a. Curiously, the visibility of MAR features reduces with increasing Little-Parks lobe, which makes it more difficult to compare the experimental data with the spectral gaps for $B \gtrsim 100$ mT. Regardless, a reasonable agreement with the data is observed (more clearly seen in the zeroth lobe), even though our experiment is not able to resolve the splitting between the $\Omega_1/e$ and $\Omega_2/e$ peaks (see also Supplementary information Fig. S2).

## Demonstration of large device variability

Applying Joule spectroscopy to a number of different samples underscores that each device is unique. We present below two addi-tional examples of devices based on nominally identical NWs. We start with device B, which has the same geometry as device A with the exception that the lengths of the epitaxial Al leads are made purpo-sefully asymmetric $(L_{1(2)} \approx 0.5(0.7)\mu m)$. The low-bias transport response shown in Fig. 4a is similar to that of device A, although the MAR oscillations with $B$ are not as clearly discernible. Despite the similarities, Joule spectroscopy reveals that this device is in fact quite different. It demonstrates that one of the Al leads is not doubly con-nected, as concluded from the fact that only one of the dips displays the Little-Parks effect (Fig. 4b). Such a behavior can be linked to a discontinuity in the Al shell formed either during growth or the wet etching of the shell. Note that the different values of $V_{dip,i}$ are due to differences in $R_{lead,i}$, which scale with the lead length. In analogy to device A, we compare the information gained from the dips (shown as dashed lines in Fig. 4a) with the low-bias data. We obtain a reasonable correspondence with the experimental data, including the splitting between the $\Omega_1/e$ and $\Omega_2/e$ lines, which is particularly visible in the zeroth lobe.

In our last example, we study a device with a 4-terminal geometry and with normal (Cr/Au) electrical contacts to the Al-InAs NW (device C). $L_i$ in this device is also asymmetric (here, taken as the distance from the junction to the voltage probes). Figure 4d displays the zero-bias $dV/dI$ of the device as a function of $T$ and $B$. At $B = 0$, it is easy to identify that $dV/dI$ increases more abruptly at two given temperatures. Joule spectroscopy taken as a function of $T$ and at $B = 0$ (Fig. 4e) reveals that the two superconducting leads display different critical temperatures, $T_{c,1} \approx 1K$ and $T_{c,2} \approx 1.33K$. This behavior owes to the inverse super-conducting proximity, which scales inversely with the distance to the

Cr/Au contacts. In analogy to device A, we fit $V_{dip,i}(B)$ with AG theory (Supplementary information Fig. 2), and use the same fitting parameters to obtain $T_{c,i}(B)$, which are plotted as dashed lines in Fig. 4d. As in the previous examples, a very good agreement is obtained with the experimental data.

## Discussion

To conclude, we have demonstrated that the Joule effect can be fostered to provide a quick and detailed fingerprint of hybrid superconductor-semiconductor devices. By studying nominally-identical Al-InAs nanowires, we observe that intrinsic disorder and defects in the epitaxial shell, and extrinsic factors, such as the inverse superconducting proximity effect, inevitably contribute to making each device unique. Concretely, this results in asymmetries in the superconducting leads that often remain undetected owing to the difficulty to obtain separate information from the individual leads in low-bias measurements. We have shown that these asymmetries can be substantial, directly impacting the device response and that they can be further amplified with external magnetic fields, a regime which has been largely explored in the past decade in the context of topological superconductivity[35]. Joule spectroscopy thus constitutes a powerful complementary tool to low-bias transport. Clearly, the technique is not restricted to the material platform investigated here, and will also be of use for the characterization of novel materials[36–38]. Our work also points out the importance of heating in hybrid superconducting devices. Indeed, owing to the poor thermal conductivity of superconductors, the device temperature can be considerable even at voltages way below the superconductor-to-normal metal transitions discussed here, and possibly also in microwave experiments which are currently carried out in these devices[6–8]. To the best of our knowledge, such heating effects have not been typically taken into account in this type of systems. Future work is needed to further clarify heat dissipation mechanisms, e.g., by studying devices with suspended nanowires, and to evaluate possible consequences of heating in device response.

## Methods

### Sample fabrication and measured samples

The devices studied in this work are based on InAs nanowires (nominal diameter, $d = 135$ nm) fully covered by an epitaxial Al shell (nominal thickness, $t = 20$ nm). The nanowires are deterministically transferred from the growth chip to Si/SiO$_2$ (300 nm) substrates using a micro-manipulator. E-beam lithography (EBL) is then used to define a window for wet etching an approx. 200 nm-long segment of the Al shell. A 30 s descumming by oxygen plasma at 200 W is performed before immersing the sample in the AZ326 MIF developer (containing 2.38% tetra-methylammonium hydroxide, TMAH) for 65 s at room temperature. Electrical contacts and side gates are subsequently fabricated by standard EBL techniques, followed by ion milling to remove the oxide of the Al shell, and metallization by e-beam evaporation at pressures of ∼$10^{-8}$ mbar. Here, we have explored devices with two different types of electrical contacts, namely superconducting Ti (2.5 nm)/Al (240 nm) or normal Cr (2.5 nm)/Au (80 nm), the latter of which were deposited by angle evaporation to ensure the continuity of the metallic films.

Overall, we have measured a total of 18 devices from 6 different samples. The main features discussed in this work have been observed in all of the devices. We focus our discussion in the main text to data corresponding to three devices from three different samples. Device A was fabricated with superconducting Ti/Al contacts and a side gate approximately 100 nm away from the junction. The nominal lengths of its epitaxial superconducting leads were $L_1 = 0.42\,\mu$m, $L_2 = 0.45\,\mu$m. Device B also had superconducting Ti/Al contacts, but the charge carrier density was tuned by a global back gate (here, the degenerately-doped Si substrate, which is covered by a 300 nm-thick SiO$_2$ layer). The lengths of the epitaxial superconducting leads were made

purposefully asymmetric (nominal lengths $L_1 = 0.5\,\mu$m, $L_2 = 0.7\,\mu$m) to further confirm the impact of $R_{lead,i}$ on $V_{dip,i}$. Finally, device C had a four-terminal geometry with normal Cr/Au contacts and a global back gate. The lengths of the epitaxial leads (in this case, the distance from the junction to the voltage probes) were nominally $L_1 = 0.3\,\mu$m, $L_2 = 0.6\,\mu$m.

### Measurements

Our experiments were carried out using two different cryogenic systems: a $^3$He insert with a base temperature of 250 mK, employed in the measurements of devices A and C, and a dilution refrigerator with a base temperature of 10 mK, which was used in the measurements of device B. The DC wiring of the former (latter) consisted of pi filters at room temperature, constantan (phosphor bronze) twisted pairs down to the $^3$He pot (mixing chamber), followed by low-temperature RC filters with a cut-off frequency of 10 kHz. For the lines of the dilution refrigerator, we additionally installed low-pass filters with cut-off frequencies of 80 MHz, 1450 MHz and 5000 MHz at the level of the mixing chamber. $T_{bath}$ was measured by RuO$_2$ thermometers attached to the $^3$He pot and the mixing chamber of the above systems.

We have performed both voltage-bias (devices A and B) and current-bias (devices A and C) transport measurements using standard lock-in techniques. Typically, for a given device, we have taken different measurements both at "low-bias" and "high-bias". The former refers to limiting $V$ and $I$ to focus on the Josephson and Andreev transport that occurs for $V \leq 2\Delta/e$. By contrast, the latter corresponds to biasing the device enough to reach the regime whereby Joule effects become significant. We have employed different levels of lock-in excitation for the "low-bias" and "high-bias" measurements. Respectively, the lock-in excitations were: $dV = 5$–$25\,\mu$V and $dV = 100$–$200\,\mu$V for voltage-bias measurements (note: the $dV$ values listed above are nominal, i.e., without subtracting the voltage drop on the cryogenic filters), and $dI = 2.5$ nA and $dI = 20$ nA for current-bias measurements.

### Data processing

The voltage drop on the total series resistance of two-terminal devices (devices A and B), which are primarily due to cryogenic filters (2.5 kΩ per experimental line), have been subtracted for plotting the data shown in Figs. 1, 2 and 4a, b.

### Data analysis

Following previous work on full-shell Al–InAs nanowires[29,31], we employ a hollow cylinder model for the Al shell, assumed to be in the dirty limit, which is justified by the fact that the electron gas in Al–InAs hybrids accumulates at the metal-superconductor interface. In this geometry, the application of a parallel magnetic field leads to an oscillating pair-breaking parameter[39],

$$\alpha_\parallel = \frac{4\xi_S^2 T_c(0)}{A}\left[\left(n - \frac{\Phi_\parallel}{\Phi_0}\right)^2 + \frac{t_S^2}{d^2}\left(\frac{\Phi_\parallel^2}{\Phi_0^2} + \frac{n^2}{3}\right)\right], \quad (5)$$

with $n$ denoting the fluxoid quantum number, $A$ the cross-sectional area of the wire, $t_S$ the thickness of the Al shell, and $\Phi_\parallel = B_\parallel A$ the applied flux. For a perpendicular field, a monotone increase of pair-breaking is observed (see Supplementary information Fig. S3), which we fit to the formula of a solid wire assuming $d \lesssim \xi_S$ with $d$ denoting diameter[30,31,39],

$$\alpha_\perp = \frac{4\xi_S^2 T_c(0)\lambda}{A}\frac{\Phi_\perp^2}{\Phi_0^2}, \quad (6)$$

with $\Phi_\perp = B_\perp A$ and $\lambda$ being a fitting parameter[31]. In our analysis of parallel fields we include a small angle, $\theta$, between the external field and the nanowire axis, which is typically present in the experimental setup (see Fig. 1a). This angle is treated as a fitting parameter and can be distinct between lead 1 and 2 due to a possible curvature of the NW.

Consequently, the full pair-breaking is given by $\alpha(B) = \alpha_\parallel(B) + \alpha_\perp(B)$ with $B_\parallel = B \cos\theta$ and $B_\perp = B \sin\theta$ from which we can extract the critical temperature, $T_c(\alpha)$, using AG theory,

$$\ln\left(\frac{T_c(\alpha)}{T_c(0)}\right) = \Psi\left(\frac{1}{2}\right) - \Psi\left(\frac{1}{2} + \frac{\alpha}{2\pi k_B T_c(\alpha)}\right), \qquad (7)$$

where $\Psi$ is the digamma function. From the proportionality, $T_c(B)/T_c(0) \approx V_{dip}(B)/V_{dip}(0)$, we obtain good fits for all devices and leads assuming $t_S \approx 15$ nm (see SI), close to the nominal thickness of 20 nm from the crystal growth. This discrepancy is attributed to uncertainties in the Al deposition thickness during growth, and to the formation of an oxide layer present on all shells. From these fits we obtain the coherence lengths, $\xi_{S,i}$, and find distinct values for lead 1 and 2 in all devices. We note that the obtained $\xi_{S,i}$ values are in good agreement with values estimated from the mean-free path of the Al shell. From LP periodicity we extract wire diameter and find $d_A$, $d_C \approx 125$ nm and $d_B \approx 105$ nm with $A$, $B$ and $C$ indicating device. For these values $d_i \gtrsim \xi_{S,i}$, possibly leading to slight modifications of Eq. (6) which are accounted for by the fitting parameter $\lambda$. The discrepancy between the estimated values for devices $A$ and $C$ with respect to the nominal diameter is attributed to the diameter distribution obtained in the employed growth conditions. The thinner wire in device $B$, on the other hand, could result from special growth conditions (i.e., by sharing some of the substrate adatom collection area with a spurious extra wire). Further details and tables of device parameters can be found in the Supplementary Information.

For finite magnetic fields, the linear BCS relation between $T_c(B)$ and $\Delta(B)$ is no longer valid. Our theoretical simulations indicate that in this limit, the MAR features follow the spectral gap, $\Omega(B) \approx \Delta_0(T_c(B)/T_c(0))^{5/2}$ (see SI). This relation is used to fit low-bias MAR signatures from high-bias measurements of $V_{dip}$.

## Data availability
The datasets generated during and/or analyzed during the current study are available from the corresponding author upon reasonable request.

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

## Acknowledgements

The authors wish to thank Marcelo Goffman, Hughes Pothier, Cristian Urbina, Ramón Aguado and Elsa Prada for useful comments. We acknowledge funding by EU through the European Research Council (ERC) Starting Grant agreement 716559 (TOPOQDot), the FET-Open contract AndQC, by the Danish National Research Foundation, Innovation Fund Denmark, the Carlsberg Foundation, and by the Spanish AEI through Grant No. PID2020-117671GB-I00 and through the "María de Maeztu" Programme for Units of Excellence in R&D (CEX2018-000805-M) and the "Ramón y Cajal" programme grant RYC-2015-17973.

## Author contributions

A.I. fabricated the device, A.I., M.G., and E.J.H.L. performed the measurements and analyzed the experimental data. G.O.S. and A.L.Y. developed the theory. G.O.S. performed the theoretical calculations. T.K. and J.N. developed the nanowires. All authors discussed the results. A.I., M.G., G.O.S., A.L.Y., and E.J.H.L., wrote the manuscript with input from all authors. E.J.H.L. proposed and guided the experiment.

## Competing interests

The authors declare no competing interests.
