## [Peer Review File · Nature Communications]

Joule spectroscopy of hybrid superconductor-semiconductor nanodevicesREVIEWER COMMENTS

Reviewer #1 (Remarks to the Author):

The hybrid semiconductor-superconductor nano-structures have been foreseen to build topologically protected qubits for quantum computing. Therefore, it indeed attracted intensive attention in recent years. The authors proposed a new method to characterize such nano-structures by using of a thermal signature in the differential conductance measurement above the superconducting gap. The authors then observed the Little-Park effect in the Al shell of the leads from the thermal signature. The modulation details of the Little-Park effect are proposed to be a characterization tools for the Al shelled InAs nanowires.

The author found a unique perspective of such a nano-structure, and prove it can be employed to examine the quality of The hybrid semiconductor-superconductor nano-structures. The manuscript overall is very well written, and the experiment details are thorough. I strongly suggest it to be published in Nature Communications. I only have a few minor comments and questions to the authors.

1. A SEM picture of the sample should be included. The authors stated that there is a curvature in the nanowires. But the cartoon gives a straight line. A real picture would be helpful.
2. Fig4d should be better to include the graph of $V_{\text{dip},i}$ as a function of B since V_{dip} is the key to the whole manuscript.
3. Why the noise in Fig4b are suddenly increased?
4. The authors use Abrikosov-Gor'kov theory to fit the Little-Park modulation, and attribute the difference of the modulation to the superconducting coherence length. Is it possible that the difference come from other parameters like the thickness of the shell or the angle of the magnetic field? The authors should include the discussion in the manuscript.

Reviewer #2 (Remarks to the Author):

Being one fundamental building block for realizing topological superconductivity and superconducting qubit, great efforts have been made to improve superconductor-semiconductor nanodevices, and yet one of the bottleneck is still defects and disorders in these nanodevices. This work uses Joule heating as a spectroscopy to characterize the basic properties of the superconductor-semiconductor nanodevices, such as the lead resistance, superconducting coherent length, inverse superconducting proximity effect, critical temperature, etc. Rather than most works that focus on the low bias region, the paper focuses on the high-bias region which has been more often overlooked. From my point of view, the paper is comprehensive and the discussion is solid. I'm sure it is helpful for the field as a basic and easy-to-adapt tool. I would like to recommend publication in Nature Communications after the authors address my following questions.

1. The most basic assumption of the model is that the all the heat dissipation happen through quasiparticles in the leads. The authors estimated the e-ph coupling to be weak in Al shell, but I wonder if the substrate can also serve as a dissipation path. Is it experimentally possible to fabricate suspended nanowire for ruling out this effect? Although this simple model the authors used agreed surprisingly well with their experiment, it is still important to scrutinize more on the validity of the assumption for follow-up works in the field.
2. How is the temperature measured in the system? Did the authors use on-chip thermometer?
3. The authors mention about the asymmetry with respect to the sign of bias in Fig. 2a, and it is even more pronounced in Fig. 4b. Is it expected to be an intrinsic or extrinsic observation,

or is there any explanation on this? In the same time, I wonder if the Joule spectroscopy can distinguish the information from either side of the Josephson junction by itself, and if the asymmetry of spectroscopy tells any information on that.

5. It would be helpful if the authors can give a more intuitive way to understand that power law coefficient γ being 3.4 agrees with the mechanism of quasiparticle dissipation.

6. The authors state that AG fitting reveals that the different V_{dip} oscillations as a function of B field is due to different coherent lengths. Since the derived equation of V_{dip} includes lead resistance, I'm wondering why geometric factor such as wire length does not come into play here.

Reply to reviewers, “Joule spectroscopy of hybrid superconductor-semiconductor nanodevices” by Ibabe et al.

We thank the reviewers for their careful evaluation of our manuscript and for their positive feedback. In the following, we provide a detailed point by point answer to their specific comments and questions, and present the changes made in the manuscript in response to the reports.

Reviewer # 1

1. A SEM picture of the sample should be included. The authors stated that there is a curvature in the nanowires. But the cartoon gives a straight line. A real picture would be helpful.

The SEM picture below has been added as an Extended Data Figure (in the revised version of the manuscript, Extended Data Figure 1). It depicts a device with a similar geometry to that employed for the devices discussed in our manuscript. We have also added the following sentence in the main text (page 2):

“An electron micrograph of a typical device is shown in Extended Data Fig. 1.”

Full-shell Al-InAs junctions. Electron micrograph (false color) of a device that is lithographically similar to the devices studied in the main text.

By taking a closer look at the micrograph (see zoomed in image below), it is possible to observe that the nanowire leads display different angles, $\theta_1 \approx 2.6^\circ$ and $\theta_2 \approx 4.5^\circ$, with respect to the horizontal axis, thus supporting our assumption in the AG fitting that the wires display some intrinsic curvature.

2. Fig4d should be better to include the graph of $V_{\text{dip},i}$ as a function of B since V_{dip} is the key to the whole manuscript.

We have opted to include a measurement of $T_{\text{c},i}$ as a function of the B field in Fig. 4d, because it is a more striking demonstration of the inverse superconducting proximity effect in device C. By contrast, the $V_{\text{dip},i}(B)$ measurement, which is shown as an Extended Data Figure (Extended Data Fig. 2 in the revised version of the manuscript), is visually not very different from those corresponding to devices A and B. The main reason for this is the fact that $V_{\text{dip},i}$ depends both on the normal resistance of the lead, $R_{\text{lead},i}$, and its superconducting critical temperature, $T_{\text{c},i}$. As such, we are unable to observe that the two leads show different $T_{\text{c},i}$ directly from the $V_{\text{dip},i}(B)$ measurement. This conclusion is reached by analyzing the temperature dependence of the dips, shown in Fig. 4e. Due to the reasons mentioned above, we still prefer to keep Fig. 4d as it is. We emphasize that the $V_{\text{dip},i}(B)$ data requested by the reviewer is shown in an Extended Data Figure, along with the experimental data from all three devices.

3. Why the noise in Fig4b are suddenly increased?

We note that the measurement shown in Fig. 4b corresponds to a different device (device B) when compared to the data discussed in Figs. 1-3 (device A). After measuring a large number of devices (18

in total, as mentioned in the Methods section), we conclude that the increased noise observed in Fig. 4b is device dependent and could be indicative of disorder and of charge fluctuations around the nanowire.

4. The authors use Abrikosov-Gor'kov theory to fit the Little-Park modulation, and attribute the difference of the modulation to the superconducting coherence length. Is it possible that the difference come from other parameters like the thickness of the shell or the angle of the magnetic field? The authors should include the discussion in the manuscript.

In the Supplementary Information section S1.C 'Determining device parameters', we investigated in detail the constraints of possible fits and found that the effect of shell-thickness, t_S , and angle, θ , mostly affects the lobe decay characterized by C_2 . Below, we present a short summary of the discussion contained therein:

The simple explanation for which t_S and θ do not have the same effect as the coherence length in the fitting goes as follows:

- (i) Knowing λ from a perpendicular fit and measuring the lobe decay, C_2 , constraints θ to small values as else t_S would have to cross zero and become imaginary to fit the measured C_2 .
- (ii) Measuring the LP lobe periodicity, $B_p = \Phi_0/A \cos(\theta)$, renders area, A , largely insensitive to changes of t_S and θ as the cosine is approximately constant for small angles.
- (iii) From measuring the amplitude of LP lobes, $C_1 = 4 \xi_S^2 T_c(0) / A$, we find ξ_S from knowing $T_c(0)$ and A , and as A only weakly depends on t_S and θ so does ξ_S .

To summarize, it is not possible to fit the different amplitudes of LP oscillations from the two dips by considering that they display the same superconducting coherence length. As the above discussion is rather technical, we have opted to keep it in the Supplementary Information.

Reviewer # 2

1. The most basic assumption of the model is that the all the heat dissipation happen through quasiparticles in the leads. The authors estimated the e-ph coupling to be weak in Al shell, but I wonder if the substrate can also serve as a dissipation path. Is it experimentally possible to fabricate suspended nanowire for ruling out this effect? Although this simple model the authors used agreed surprisingly well with their experiment, it is still important to scrutinize more on the validity of the assumption for follow-up works in the field.

First, we would like to note that, as mentioned by the reviewer, our model is able to describe extremely well our experimental results, thus indicating that quasiparticle diffusion is the dominant heat dissipation mechanism. We agree with the reviewer that for future works addressing heat dissipation in hybrid devices in detail, it would be important to evaluate the contribution of other dissipation mechanisms. This, however, was out of the scope of the present work. We also agree that towards the above goal, it would be interesting to study devices with suspended nanowires. For this

work, we did not attempt to suspend our nanowires, although it is in principle experimentally feasible. We have modified the last sentence of the conclusion, to consider this interesting avenue of research for follow-up works:

“Future work is needed to further clarify heat dissipation mechanisms, e.g., by studying devices with suspended nanowires, and to evaluate possible consequences of heating in device response.”

To finalize, we would like to comment that previous work on heat effects in metals (e.g., Wellstood et al., PRB 49, 5942 (1994)) mentions that the thermal (Kapitza) resistance between a substrate and very thin films is expected to be small, as the phonon distribution in the film cannot be considered separate from that of the substrate. From this, we interpret that dissipation through the substrate would be limited by the electron-phonon coupling, which we estimate to be weak. Without a doubt, measurements in suspended wires would allow to verify this experimentally.

2. How is the temperature measured in the system? Did the authors use on-chip thermometer?

As mentioned in the “Methods” section, the measurements were taken in two different cryogenic systems: a 3He insert and a dilution refrigerator. The values of the bath temperature (T_{bath}) mentioned in the manuscript were obtained by thermometers attached to the 3He pot and the mixing chamber of the above systems (specifically, RuO₂ thermometers). We have added the following sentence to the “measurements” sub-section in Methods (page 7):

“ T_{bath} was measured by RuO₂ thermometers attached to the 3He pot and the mixing chamber of the above systems”.

For this work, we have not employed on-chip thermometers. This would be clearly interesting for future studies, as it would allow to probe the temperature in different parts of the device as Joule heating takes place.

3. The authors mention about the asymmetry with respect to the sign of bias in Fig. 2a, and it is even more pronounced in Fig. 4b. Is it expected to be an intrinsic or extrinsic observation, or is there any explanation on this? In the same time, I wonder if the Joule spectroscopy can distinguish the information from either side of the Josephson junction by itself, and if the asymmetry of spectroscopy tells any information on that.

Our experimental data indicates that the asymmetry mentioned by the reviewer is intrinsic to the devices. To reach this conclusion, we have first verified that the dip asymmetry is not affected by the sweep direction of the applied voltage (panel a in the figure below). This rules out heating as the origin of the asymmetry, as the same dip structure is observed regardless of the measurement being taken from higher to lower temperature (from finite voltage to zero) or from lower to higher temperature. By contrast, we observe that the dip structure “flips” with respect to the sign of the applied voltage when we swap the contacts to which the voltage is applied and from which the current is measured (see panel b in the figure below). This indicates that the effect is intrinsic to asymmetries of the leads in the device.

Dip asymmetry with respect to the sign of the voltage bias in device A ($V_g = 80$ V). **a**, $dI/dV(V)$ curves taken for different sweep directions of the voltage bias, V . The asymmetry is not affected by the sweep direction, which rules out heating as the underlying effect. **b**, $dI/dV(V)$ traces taken by inverting source and drain contacts (see inset for the different measurement configurations). The dip asymmetry flips with respect to the sign of the voltage bias, indicating that it is an intrinsic effect related to the asymmetry of the leads.

A possible source for the above asymmetry could be related to weak non-linearities in the lead resistances, $R_{\text{lead},i}$, assuming that they depend weakly on the chemical potential, μ , of the wires. For positive voltage one side experiences higher μ and the other lower. Reversing voltage reverses the chemical potential difference. This in effect would move $P_{\text{dip},i}$ slightly by changing $R_{\text{lead},i}$ in reversing biases, while not affecting measured dI/dV which is determined by the junction. Such non-linearities could arise from mesoscopic fluctuations in the proximitized semiconductor or due to the presence of charge traps.

5. It would be helpful if the authors can give a more intuitive way to understand that power law coefficient γ being 3.4 agrees with the mechanism of quasiparticle dissipation.

As expected for a BCS superconductor, thermal conductivity is exponentially suppressed at low temperatures and scales linearly with T close to T_c . This behavior is responsible for the temperature dependence of the power at which the superconductor-to-normal transition of the leads takes place. The γ value 3.6 gives a very good fit to the numerically calculated dip power, P_{dip} as a function of T_{bath} (see Supplementary Information, sections S2A and S2B). This value is a compromise, attempting to bridge the exponential suppressed dependence for small T_{bath} to the

second order closing, $\gamma \approx 2$, for T_{bath} close to T_C . To make it clearer, we have added the following comment in page 5 of the main text:

“As shown in the Supplementary Information, we numerically calculate P_{dip} as a function of T_{bath} and fit the resulting curve to eq. 4, obtaining $\gamma_{\text{theory}} \approx 3.6$, which is in excellent agreement with our experimental results.”

6. The authors state that AG fitting reveals that the different V_{dip} oscillations as a function of B field is due to different coherent lengths. Since the derived equation of V_{dip} includes lead resistance, I'm wondering why geometric factor such as wire length does not come into play here.

As the reviewer correctly pointed out, $V_{\text{dip},i}$ depends both on the normal state resistance of the lead, $R_{\text{lead},i}$, and on its superconducting critical temperature, $T_{c,i}$ (the latter can be tuned by the applied magnetic field). Our observations indicate that $R_{\text{lead},i}$ does depend on geometrical factors, such as the length of the leads, in agreement with the reviewer's expectation. To verify this, we have made devices B and C purposefully asymmetric with respect to the length of the leads. Accordingly, a larger difference between the values of $R_{\text{lead},1}$ and $R_{\text{lead},2}$ is obtained in these devices (see Tables S1, S2 and S3 in the Supplementary Information).

We now turn to the measurements taken as a function of the B field. For the AG fitting, we consider that $R_{\text{lead},i}$ is not affected by the magnetic field. Note that $R_{\text{lead},i}$ englobes all the geometrical factors of the lead, and hence, it does not play a role in the response of the dips with the B field. For devices with normal metal lithographic contacts (Cr/Au), such as device C, the $R_{\text{lead},i}$ values used for the AG fitting are the same as those obtained from fits of $V_{\text{dip},i}$ as a function of R_J at zero magnetic field (see Extended Data Fig. 3, or Extended Data Fig. 4 in the revised version of the manuscript). For devices with superconducting lithographic contacts (Ti/Al, devices A and B), $R_{\text{lead},i}$ estimated from measurements at $B = 0$ is approx. 10% larger than the value obtained from the AG fitting. We attribute this to an increased higher thermal resistance of the device at zero field due to the opening of the superconducting gap of the lithographic contacts (as discussed in section S1B of the Supplementary Information). Finally, we note that the excellent agreement between the fits of $V_{\text{dip},i}(B)$ and $T_{c,i}(B)$, using the same AG parameters, support our assumption that $R_{\text{lead},i}$ is not affected by the B field (see e.g., Figs. 3a and 3c in the main text).

REVIEWERS' COMMENTS

Reviewer #1 (Remarks to the Author):

The authors have answered all my questions. I have no more further questions. The manuscript is ready to be published.

Reviewer #2 (Remarks to the Author):

The authors have addressed my questions satisfactorily. I have one additional comment on the temperature sensing part. If the sample is cooled down mainly by wiring, I would expect at low temperatures (below a few hundreds millikelvin) there can be a large temperature mismatch between the nominal temperature on He3 pot/mixing chamber and the sample. This could be affected by filtering/wiring material/contact leads material. Although they don't use an on-chip thermometer, it will be nice if the authors can comment or estimate the accuracy of their temperature measurement, as this constitutes one of their main fitting results. I would like to recommend publication on Nature Communications after the authors comment on this part.

Reply to reviewers, "Joule spectroscopy of hybrid superconductor-semiconductor nanodevices" by Ibabe et al.

We thank again the reviewers for their positive feedback on our manuscript. In the following, we comment on the final point raised by reviewer #2.

Reviewer #2 (Remarks to the Author):

The authors have addressed my questions satisfactorily. I have one additional comment on the temperature sensing part. If the sample is cooled down mainly by wiring, I would expect at low temperatures (below a few hundreds millikelvin) there can be a large temperature mismatch between the nominal temperature on He3 pot/mixing chamber and the sample. This could be affected by filtering/wiring material/contact leads material. Although they don't use an on-chip thermometer, it will be nice if the authors can comment or estimate the accuracy of their temperature measurement, as this constitutes one of their main fitting results. I would like to recommend publication on Nature Communications after the authors comment on this part.

We agree with the referee that the electron temperature of a sample and the nominal temperature of a cryostat can be different when the sample is cooled mainly by wiring. As also mentioned, these differences are strongly impacted by the wiring/filtering of the experiment and become more important for very low temperatures.

The great majority of data presented in our manuscript (Figs. 1-3 and Fig. 4c-d, corresponding to devices A and C) was taken using a ^3He insert with a base temperature of approx. 250 mK. The DC wiring in this system consisted of room temperature pi filters, constantan twisted pairs down to the ^3He pot, followed by low-temperature two-stage low-pass (RC) filters with a cut-off frequency of 10 kHz and a resistance of 2.5 k Ω . Overall, this wiring is very similar to that described by, e.g., Torresani et al., Phys. Rev. B 88, 245304 (2013), who showed a near one-to-one correspondence between electron and cryostat temperatures down to ~ 300 mK (as concluded by fitting Coulomb blockade peaks of a quantum dot confined in a GaAs-based 2DEG). Owing to the similarities of our setup, we expect the electron temperature in our experiment to be comparable to those in the reference above. To obtain an estimate, we have analyzed Coulomb blockade peaks of a quantum dot formed spontaneously in an InAs nanowire. Due to the simplicity of the studied device geometry (with a single gate), we were unable to finely tune the dot to the weak coupling limit where the peak width is limited by temperature. As such, our analysis is only able to extract an upper bound estimate of the electron temperature. For measurements taken nominally at 250 mK, we obtain an upper

bound of approx. 400 mK. We note nonetheless that the good agreement of the Abrikosov-Gork'ov fit to the Little-Parks oscillations of Fig. 3c (down to cryostat temperatures of 250 mK) suggests that the temperature of the sample does not saturate at 400 mK. This thus indicates that the electron temperature is lower than the upper bound estimate above.

We note that the mismatch between sample and cryostat temperature will be larger for measurements taken in a dilution refrigerator (device B, nominal temperatures down to 10 mK). For this setup, the twisted pairs consisted of phosphor bronze (rather than constantan), and we have installed additional high frequency low-pass filters to the DC lines at the level of the mixing chamber with cut-off frequencies of 80 MHz, 1450 MHz and 5000 MHz.

Despite not having a very precise measurement of the sample temperature at the chip level, we emphasize that the main conclusions of our work are not very sensitive to T_{bath} , as long as it does not reach temperatures of the order of T_{c} . Indeed, as we discuss in eq. (3) in the main text and in Fig. S8, the proportionality between V_{dip} and T_{c} relies on the prefactor Λ remaining approximately constant within the experimental parameter space, which is the case up to large depairing ($\alpha \sim 0.4$) even for finite temperature (Fig. S8c).

Of course, as we mentioned in our previous reply, we fully agree with the reviewer that having on-chip thermometers would open the way for exploring the physics of Joule heating and of the heat dissipation mechanisms in hybrid devices in greater detail. This is definitely worth pursuing in future experiments.

We have added the following description of our DC line filtering to the Methods section:

"The DC wiring of the former (latter) consisted of pi filters at room temperature, constantan (phosphor bronze) twisted pairs down to the 3He pot (mixing chamber), followed by low-temperature RC filters with a cut-off frequency of 10 kHz. For the lines of the dilution refrigerator, we additionally installed low-pass filters with cut-off frequencies of 80 MHz, 1450 MHz and 5000 MHz at the level of the mixing chamber."